# Positive Association of Dietary Inflammatory Index with Incidence of Cardiovascular Disease: Findings from a Korean Population-Based Prospective Study

**DOI:** 10.3390/nu12020588

**Published:** 2020-02-24

**Authors:** Imran Khan, Minji Kwon, Nitin Shivappa, James R. Hébert, Mi Kyung Kim

**Affiliations:** 1Cancer Epidemiology Branch, Division of Cancer Epidemiology and Prevention, National Cancer Center, Goyang 10408, Gyeonggi-do, Korea; imrankhan572@yahoo.com (I.K.); 74433@ncc.re.kr (M.K.); 2Cancer Prevention and Control Program, University of South Carolina, Columbia, SC 29208, USA; shivappa@email.sc.edu (N.S.); jhebert@mailbox.sc.edu (J.R.H.); 3Department of Epidemiology and Biostatistics, Arnold School of Public Health, University of South Carolina, Columbia, SC 29208, USA; 4Department of Nutrition, Connecting Health Innovations LLC, Columbia, SC 29201, USA

**Keywords:** cardiovascular diseases, chronic inflammation, dietary inflammatory index, KoGES cohort, Korean, myocardial infarction, stroke

## Abstract

Recently, diets with higher inflammatory potentials based on the dietary inflammatory index (DII^®^) have been shown to be associated with increased cardiovascular disease (CVD) risk in the general population. We aimed to prospectively investigate the association between the DII and CVD risk in the large Korean Genome and Epidemiology Study_Health Examination (KoGES_HEXA) cohort comprised of 162,773 participants (men 55,070; women 107,703). A validated semi-quantitative food frequency questionnaire (SQ-FFQ) was used to calculate the DII score. Statistical analyses were performed by using a multivariable Cox proportional hazard model. During the mean follow-up of 7.4 years, 1111 cases of CVD were diagnosed. Higher DII score was associated with increased risk of CVD in men (hazard ratio [HR]_Quintile 5 vs. 1_ 1.43; 95% CI 1.04–1.96) and in women (HR_Quintile 5 vs. 1_ 1.19; 95% CI 0.85–1.67), although not significant for women. The risk of CVD was significantly higher in physically inactive men (HR_Quintile 5 vs. 1_ 1.80; 95% CI 1.03–3.12), obese men (HR_Quintile 5 vs. 1_ 1.77; 95% CI 1.13–2.76) and men who smoked (HR_Quintile 5 vs. 1_ 1.60; 95% CI 1.10–2.33), respectively. The risk of developing stroke was significantly higher for men (HR_Quintile 5 vs. 1_ 2.06; 95% CI 1.07–3.98; *p* = 0.003), but not for women. A pro-inflammatory diet, as indicated by higher DII scores, was associated with increased risk of CVD and stroke among men.

## 1. Introduction

Diet has been known to play an important role in modulating chronic inflammation [1,2,3]. Chronic inflammation is characterized by the constant presence in the blood stream of pro-inflammatory cytokines that are associated with tissue injury as a result of histamine produced by damaged mast cells [4]. Inflammation is caused as a response to repeated injury and involves the release of pro-inflammatory cytokines including tumor necrosis factor-α (TNF-α), interleukin (IL)-1, and IL-6 [5], which promotes the progression of atherosclerosis, leading to plaque rupture and thrombosis [4]. The atherothrombotic events lead to the development of cardiovascular diseases (CVD), especially its subtypes myocardial infarction (MI) and most strokes [6,7]. Data from the United States alone suggest that approximately 0.55 million first episodes and 0.2 million repeated episodes of acute MI occur annually [8], while 15 million people suffer from stroke worldwide with five million deaths and leaving another five million permanently disabled annually [9]. Together, MI and stroke have affected large numbers of people around the world [10,11,12] including Korea [13].

The dietary inflammatory index (DII^®^), a literature-derived tool, was developed to evaluate the inflammatory potential of an individual’s diet [14]. A higher DII score indicates a more pro-inflammatory diet, whereas a lower DII score indicates an anti-inflammatory diet [15]. An association between the DII and the inflammatory biomarkers have been reported in 25 studies [15,16,17,18]. While most articles that have examined the relationship between the DII and CVD outcomes have detected a positive association [18,19,20,21], some have not [22]. Therefore, there is a need to deepen our understanding of whether a more pro-inflammatory diet, as measured by DII score, could be associated with higher risk of CVD and its subtypes, MI and strokes.

The relationship between DII and CVD have been investigated in different cohorts and have suggested an association between DII scores and CVD risk [20,23] and MI in particular [24], though null associations have also been reported [25,26]. For example, no significant association between the DII and stroke has been reported in studies in France and Australia [24,25]. In several case-control studies, a positive association between DII and risk of MI was reported [7,27]. Previously, the association between DII and CVD risk has been explored in studies in Spain, which enrolled participants who were at high risk of developing CVD [20], or highly educated Spanish [23] and French individuals [24]. Therefore, because of the possible selection bias, these results might not be simply generalized to a population outside Europe. Moreover, the broader application of this approach to non-Western populations is also limited, as indicated by the relative lack of studies conducted in Asian populations that have addressed the association between the DII and CVD risk [26,28,29]. For example, Asadi et al. [26] recently reported the association between DII and CVD and its subtypes, MI, stable angina, and unstable angina risk in northeastern Iran; however, the incidence rate was very low and regionally specific, which makes it difficult to generalize the results to the entire Iranian population, let alone outside of Iran. Another Asian study from Japan reported the association between the DII and overall CVD and stroke mortality among other diseases, but failed to address the disease incidence risk [29]. However, to our knowledge, no previous study has been conducted to examine the association between the DII and incidence of CVD in an East-Asian population whose dietary habits are different from those of the other populations studied.

Against background, we aimed to prospectively investigate the association between the inflammatory potential of the diet, as measured by the DII, and the incidence of overall CVD and its subtypes MI and strokes in a large population-based Korean Genome and Epidemiology Study_Health Examination (KoGES_HEXA) cohort.

## 2. Materials and Methods

### 2.1. Cohort Characteristics

KoGES cohort data were used for the current study. The KoGES cohort is based on the general Korean population, and aimed to obtain information on the genetic, environmental, and lifestyle determinants (i.e., hypertension, hyperuricemia, osteoporosis, metabolic syndrome, obesity, cardiovascular disease, cerebrovascular accidents, and cancer). KoGES collects and examines biological data such as urine, blood, and genetic material as well as epidemiological survey data. KoGES includes several cohorts including the KoGES cardiovascular disease association study (CAVAS), KoGES Ansan and Ansung study, and the KoGES health examinee study (HEXA). These population-based cohorts include both men and women participants who were recruited by the National Health Examinee Registry, and were 40–79 years of age at baseline. Detailed information on KoGES cohorts can be found elsewhere [30]. Among these cohorts, data from the KoGES_HEXA cohort were used in the present study to determine the association between the DII and CVD (overall and subtypes) risk. The HEXA cohort has been recruiting new participants, mostly from examination/medical institutions around the country. A total of 173,343 participants (59,291 men, 114,052 women) were recruited from 38 health examination centers and hospitals in Korea between 2004 and 2013. All the participants voluntarily filled out a baseline survey through letters, on-site invitation, campaign, and community conferences. Surveys targeting people visiting hospitals for a physical examination were conducted in the regions of Seoul, Incheon, Busan, Gwangju, Daegu, Anyang-si, Ulsan, Seongnam-si, Goyang-si, Chuncheon-si, Cheonan-si, Hwasun-si, and Changwon-si.

### 2.2. Study Population

A total of 173,343 participants were recruited at the baseline between 2004 and 2013 and were asked to attend follow-up visits between 2007 and 2016 (mean follow up = 7.4 years). All participants voluntarily signed an informed consent form before the study and the study protocol was officially approved by the Institutional Review Board of the National Cancer Center (IRB No. NCC2018-0164) as well as the IRBs of the institutions that had participated in KoGES. Data accumulated from the baseline to the first follow-up were used and person–years of exposure time were obtained from the baseline to the first follow-up. Data from participants who had CVD (stroke, myocardial infarction, angina, and transient-ischemic attack) at the baseline had missing data (=6296), were missing caloric intake (men <500 Kcal or >6000 Kcal; women <500 Kcal or >4000 Kcal = 4272), or had no energy data were excluded from the analyses. This resulted in a final sample size of 162,773. The outcome was defined as a diagnosis of CVD by a medical doctor, which was self-reported by the participants. A total of 1111 incident cases of overall CVD (subtypes MI = 842; stroke = 288; MI + stroke = 19) were identified after fulfilling the exclusion criteria (Figure 1).

### 2.3. Covariates

For baseline recruitment, eligible participants were asked to volunteer through on-site invitation, mailed letters, telephone calls, media campaigns, or community leader-mediated conferences. For the interview, a questionnaire was administered by trained staff. Baseline assessment included a physical examination. Participants were invited to complete periodic follow-up surveys by mail and telephone calls. The information about sociodemographic characteristics, personal and family history, physical activity, physical examinations, clinical investigations, and semi-quantitative food frequency questionnaire (SQFFQ) were obtained at the baseline and follow-up examinations using a structured questionnaire [30,31,32,33]. Both continuous and categorical variables were recorded and the details can be found elsewhere [34,35]. Regularity of physical activity was determined according to whether or not subjects participated regularly in any sports to the point of sweating. Those who did so were classified as the “regular” physical activity group, while those who did not were classified as the “irregular” physical activity group.

### 2.4. Dietary Assessment and Calculation of Dietary Inflammatory Index(DII)

The participants’ dietary intakes were recorded at the baseline using a validated SQFFQ. Detailed information on the validity of the SQFFQ is available elsewhere [31]. Based on the SQFFQ, the study participants assessed their consumption frequencies and the average amounts of 106 food items consumed over the course of one year. To measure the nutrient intake per day, the total of the values of the average serving amounts and serving frequencies was applied [32]. The frequency of consumption of each food item was calculated according to nine choices, starting from “almost never” to “more than three per day”, while portion size was estimated from three response choices such as ½ serving, one serving, and 1.5 servings, and daily nutrient intakes were estimated using a Korean food composition table [33].

Details of the updated DII are available elsewhere [14]. Briefly, a total of 1943 research articles were reviewed and scored for 45 food parameters based on their effects on the levels of inflammatory markers such as CRP, IL-4, IL-6, IL-1β, IL-10, and TNF-α. In the current study, a total of 37 parameters out of 45 were available. Anti-inflammatory parameters included magnesium, vitamin C, niacin, riboflavin, beta-carotene, vitamin E, flavonones, flavan-3-ol, ginger, isoflavones, fiber, onion, monounsaturated fatty acids (MUFAs), polyunsaturated fatty acids (PUFAs), vitamin A, thiamin, vitamin B-6, vitamin D, anthocyanidins, flavonols, flavones, pepper, garlic, alcohol, folic acid, and tea. Pro-inflammatory parameters included cholesterol, total fat, protein, carbohydrate, saturated fatty acids, vitamin B12, and total calories. The nutritional data used in the current study were assessed from the Functional Ingredients Table (Rural Development Administration), Computer Aided Nutritional Analysis (The Korean Nutrition Society), and the U.S. Department of Agriculture. A global comparison database consisting of data from eleven countries were utilized to calculate the DII scores based on each of the 45 parameters (i.e., foods, nutrients, and other food constituents). The details for the DII calculation are available elsewhere [14].

### 2.5. Statistical Analysis

The association between DII and the risk of CVD (overall and subtypes) were statistically analyzed. The DII levels were represented as quintiles, based on the cohort for whom there was no baseline CVD (overall and subtypes). Continuous and categorical variables were represented as the means with standard deviation and frequency/counts as percentages, respectively. The trends in *p* values were calculated for continuous and categorical variables using the Jonckheere–Terpstra test and Mantel–Haenszel Chi-square test, respectively. The association between the baseline DII and newly diagnosed CVD (overall and subtypes) were analyzed by using the multivariable Cox’s proportional hazard model. To confirm the assumption of proportional risk, all of the models were evaluated and deemed to be consistent with a model that included time-dependent covariates. The model was adjusted for sex (for all participants); energy intake (continuous), age (continuous), BMI (continuous), alcohol consumption (categorical); physical activity (categorical) and smoke (categorical). Stratified analyses were carried out by physical activity and BMI. The hazard ratios (HRs) were calculated with 95% confidence intervals (CIs), and the two-sided probability values were judged to be statistically significant if *p* < 0.05. We assessed heterogeneity between subgroups (men and women) using a likelihood ratio test comparing Cox’s proportional hazard models with and without interaction terms for DII and gender. All of the statistical analyses were performed with SAS^®^ 9.3 (SAS Institute, Cary, NC, USA). *p* values < 0.05 were considered as statistically significant.

## 3. Results

The baseline characteristics of the participants from the KoGES_HEXA cohort based on the DII quintiles are presented in Table 1. A total of 162,773 participant (male = 55,070; female = 107,703) with evaluable data were analyzed. DII scores from food and nutrients ranged from −7.90 to +7.11. Men had a higher median DII score (0.92) than women (0.89). As the DII score increased, the mean age of the participants increased (*p* < 0.0001), while the mean energy intake decreased (*p* < 0.0001). The mean BMI of the participants with a lower DII score was higher when compared to the higher DII score (*p* < 0.0001). The percentage of participants in the lower DII score for education attainment was higher when compared to the higher DII score participants (*p* < 0.0001). Moreover, participants with higher DII scores relative to those with lower scores had a lower income, higher smoking rate, and exercised less regularly (*p* < 0.0001). The number of post-menopausal women participants increased, while the number of married people decreased as the DII increased, respectively (*p* < 0.0001). The number of never drinkers increased, while current drinkers decreased as the DII score increased (*p* < 0.0001).

During the 7.4 years follow-up, a total of 1111 individuals (578 men and 533 women) developed CVD (842 cases of MI and 288 cases of stroke). Cox’s regression analysis exhibited significantly higher risk (*p* = 0.006) of developing overall CVD (HR_Quintile 5 vs. 1_ 1.32; 95% CI 1.05–1.67) at the baseline for pro-inflammatory quintile (=Q5), while setting the lowest DII as the reference (=Q1) after adjusting for confounding variables (gender, age, smoke, alcohol consumption, physical activity, BMI, and energy intake) (Table 2). After stratification by gender, men who had a higher DII score (=Q5) had a significantly higher risk of developing overall CVD (HR_Quintile 5 vs. 1_ 1.43; 95% CI 1.04–1.96; *p* = 0.007), while the result was not significant among women (HR_Quintile 5 vs. 1_ 1.19; 95% CI 0.85–1.67; *p* = 0.18). The *p*-heterogeneity between men and women indicated no significant difference by sex (*p* = 0.64).

The Cox’s regression analysis showed no significant risk of developing MI for all participants, or for men and women separately, after adjusting for confounding variables (gender, age, smoke, alcohol consumption, physical activity, BMI, and energy intake) (Table 3). The risk of developing stroke was significantly higher for all participants (HR_Quintile 5 vs. 1_ 1.55; 95% CI 0.99–2.44; *p* = 0.008) and for men (HR_Quintile 5 vs. 1_ 2.06; 95% CI 1.07–3.98; *p* = 0.003); however, no significant results were found for women.

Table 4 reports the stratified results for overall CVD by physical activity, smoking status, and BMI. Men who did not participate regularly in physical activity exhibited significantly higher risk of developing overall CVD (HR_Quintile 5 vs. 1_ 1.80; 95% CI 1.03–3.12; *p* = 0.03) when compared to the participants who participated regularly (HR_Quintile 5 vs. 1_ 1.30; 95% CI 0.86–1.94; *p* = 0.09). However, we did not find any significant association between DII score and overall CVD in female participants after stratified by physical activity.

The risk of developing overall CVD increased, especially among physically inactive men, obese men, and smoking men. In obese men (BMI > 25 Kg/m^2^), a significant association was observed between the DII and risk of developing overall CVD (HR_Quintile 5 vs. 1_ 1.77; 95% CI 1.13–2.76; *p* = 0.01). After stratification by smoking, the risk of overall CVD was significantly higher among all participants who smoked (HR_Quintile 5 vs. 1_ 1.52; 95% CI 1.05–2.18; *p* = 0.004). When further stratified by gender, the risk of overall CVD was higher (*p* = 0.002) in male smokers (HR_Quintile 5 vs. 1_ 1.60; 95% CI 1.10–2.33).

## 4. Discussion

In this prospective cohort study, we aimed to investigate the prospective association between dietary inflammatory potential and the risk of CVD in the Korean population. We observed that the Korean participants who had higher DII scores were at 63% higher risk of developing overall CVD, which was attenuated after multivariable-adjustment (32% higher risk of overall CVD). The association was stronger among males compared to females. The risk of overall CVD was higher among men who were overweight or obese (BMI > 25 Kg/m^2^), cigarette smokers, and physically inactive. These findings indicate that the most pro-inflammatory diet (high DII score = Q5) had a higher risk of CVD compared to those with an anti-inflammatory diet (lowest DII score = Q1). Similar results were reported by Garcia-Arellano et al. [19,20] in their prospective PREDIMED cohort (i.e., those reported results showed that a pro-inflammatory diet is associated with a higher risk of CVD among the Spanish population). In accordance with our findings, Ramallal et al. [23] reported the association between DII and CVD risk among the Spanish population in the SUN cohort, reporting results showing that a pro-inflammatory diet was associated with a significantly higher risk for developing CVD. These results suggested that inflammation—a phenomenon with numerous underlying causes triggered by a pro-inflammatory diet–is associated with CVD risk. However, in contrast to our findings, several studies have reported a null association between DII and overall CVD [25,26]. The difference between our study and that of Vissers et al. [25] could be attributed to the difference in the population studied, as they used only women participants and data on 25 of the 45 DII food parameters were available from the validated food frequency questionnaire (FFQ) that was used for calculating the DII, as compared to our 37 food parameters. Moreover, the difference in the findings between ours and that of Asadi et al. [26] could be attributed to fewer incident cases CVD (*n* = 124), and regional specificity, which might be associated with different culture and lifestyle factors, and therefore, cannot be generalized to the general population of Iran. In addition, their validated FFQ had fewer food items (65 items) compared with the FFQ used (106 items) in the present study. Therefore, it is likely that they missed some factors for estimation, which were predictive in our results.

It is important to note that stroke and myocardial infarction present a multifactorial pathogenesis, which also includes genetics in addition to lifestyle and diet. Episodes of stroke and myocardial infarction may occur across generations. However, in the present study, compared to individuals without a family history of CVDs with a higher DII, individuals with lower DII but with a family history of CVDs were not more likely to develop stroke.

Based on the hypothesis that inflammation is associated an atherothrombotic outcomes [36], we would expect a significant positive association between DII and MI or stroke [19,21,37]. In this study, we found a null association between DII and MI risk, while we did observe a positive association between the DII and stroke. The association between DII and MI risk in our results are in agreement with previous reports [25,26]. However, positive association between DII and MI was also reported in the previous studies [24,27]. In a Swedish study, a null association between DII and MI was reported for women while a significantly positive association was reported for men [7]. The possible explanation for this could be that the underlying mechanism of inflammation and its effects on the subtypes of CVD events differ by sex.

To our knowledge, this is the first report of a strong association between a more pro-inflammatory diet as measured by DII and stroke risk among men in a prospective study. Lower DII score not only reflects the anti-inflammatory potential of the diet, but that the diets are generally healthier [24]. Therefore, the finding of the study should be taken in light of studies that have reported preventive roles of overall high-quality diets with respect to stroke. For example, a study conducted by the WHO in 2005 showed that a high intake of fruits and vegetables could prevent the burden of stroke and ischemic heart disease by 19% and 31%, respectively [38]. In another study conducted by the Finnish Mobile Clinic Health Examination Survey, a total of 3932 participants including both men and women, were followed for 24 years and reported results that showed an inverse association of stroke with a high intake of citrus fruit and cruciferous vegetables [39]. In Sweden, Larsson et al. [40] conducted a prospective study on 74,961 participants who were followed for a mean 10.2 years and reported results showing that total fruit and vegetable consumption reduced the multivariable relative risk of stroke by 13%, and later reported that dietary fiber intake was inversely associated with stroke incidence [41]. The protective role of fruits and vegetables [42], and certain nutrients (potassium, vitamin K, and dietary fiber) [43,44,45] could be attributed to its ability to lower blood pressure (a major risk factor for stroke). In agreement with our finding, a null association between DII and stroke among women was reported in a previous Australian study [25]. In contrast to our findings, another study reported a null association between DII and stroke among all subjects, however, no information on different risk associations between gender [24]. In addition, they reported the results for a comparatively small study of younger individuals with a higher education level when compared to the general population. Therefore, it is hard to generalize their results to other populations.

Another interesting finding of our study is that a statistically significant association was found between the DII and CVD risk among participants who were physically inactive. When stratified further for gender, the risk of CVD for physically inactive men was significantly higher than for women. Several studies have demonstrated an inverse relationship between regular exercise and the risk of coronary heart disease, cardiac events, and death [46,47,48]. There is growing evidence that those who are physically active have a reduction in biomarkers related to chronic inflammation and, with few exceptions, the anti-inflammatory effects of exercise appear to occur irrespective of chronic diseases or age [49]. However, the mechanisms behind this are still not fully elucidated.

There is concern about the effect of the growing rate of overweight and obesity and their association with CVD, stroke, metabolic syndrome, type II diabetes mellitus, cancer, and hypertension [50]. BMI has the potential to act as a mediator between diet, low-grade chronic inflammation, and inflammation related diseases, and not just as a confounder. Body fatness produces a pro-inflammatory metabolic environment in the body, and BMI is positively associated with inflammatory markers [51,52]. It was reported earlier that obesity and overweight is positively associated with an increased risk of MI and stroke [53]. We also found an increased risk of overall CVD with a higher DII score in obese men.

Smoking is considered an important risk factor for the development of CVD like many other chronic diseases [54]. We found an increased risk of CVD with higher DII score in smoking men. Smoking profoundly affects CVD events through aggravating the atherosclerosis process by promoting it through regulating blood pressure, lipid metabolism, and increasing LDL-C level in the body. Moreover, smoking is an independent risk factor for atherosclerosis because of the high concentration of pro-oxidants and pro-inflammatory compounds that are able to increase endothelial injury and inflammation [55].

This study has some limitations that need to be pointed out. First, the diagnosis of the diseases was self-reported by participants who may underestimate the incidence with some error. Second, we used an FFQ to record the food consumption for the past year; thus, an individual may misreport her/his food intake. Additionally, we lacked information on eight food parameters to calculate a DII score that could influence the results. Despite these limitations, there are certain strengths to this study. It is the first of its kind from an Asian country, which assesses the association of the inflammatory effect of diet in terms of DII score with CVD risk. This could be utilized for important clinical recommendations, especially those addressing the trend of worsening dietary patterns worldwide. Another strength of the current study includes its prospective design, which minimizes recall bias and reverse causation. Additionally, it was adjusted for several lifestyles, demographic factors, and dietary factors as confounders. Furthermore, the population-based nature of the study cohorts increased the external validity. Finally, this study included a much larger number of Korean adults and by using the KoGES_HEXA cohort database, it is a sample that has strong internal validity; a *sine qua non* of external validity.

## 5. Conclusions

In conclusion, diets with pro-inflammatory effects, as estimated by the DII, were significantly associated with an increased risk of CVD in men; however, they were not significantly associated in women. Additionally, the association between DII and subtypes of CVD was significant only in stroke, not in MI. Physical activity, smoking, and BMI acted as mediators between DII and CVD risk. These findings suggest that inflammation is a connection between diet and CVD risk as well as the usefulness of the DII in estimating the inflammatory impact of diet. Furthermore, the results underline the importance of consuming a more anti-inflammatory diet as a strategy to lower the risk of CVD for public health recommendations and guidelines for dietary recommendations in clinical settings.

## Figures and Tables

**Figure 1 nutrients-12-00588-f001:**
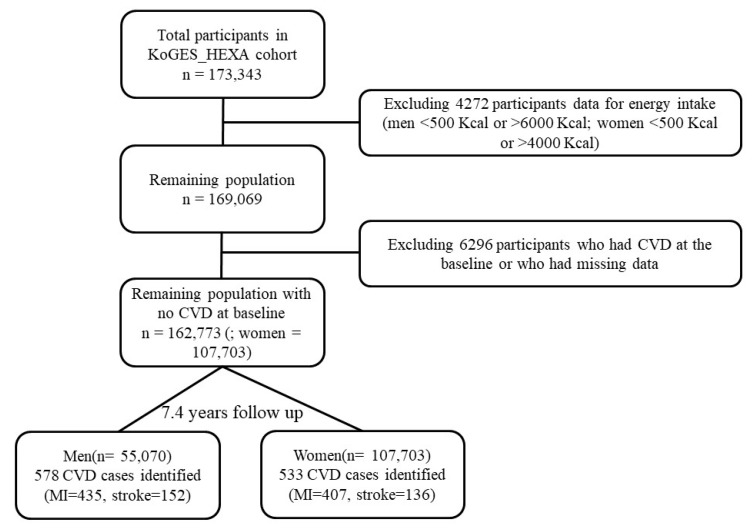
Flow chart for the sample selection from the KoGES_HEXA cohort from 2004 to 2016. CVD: cardiovascular diseases, MI: myocardial infarction.

**Table 1 nutrients-12-00588-t001:** Baseline characteristics of the study subject by dietary inflammatory index (DII) quintiles in the KOGES_HEXA cohort, 2004~2013.

Characteristics	Quintiles of DII ^a^					
	Q1 (Anti-Inflammatory)	Q2	Q3	Q4	Q5 (Pro-Inflammatory)	*p* for Trend ^b^
N (*n* = 162,773)	*n* = 32,525	*n* = 32,546	*n* = 32,548	*n* = 32,589	*n* = 32,565	
No. of cases	192	214	215	257	233	
Range ^c^	−7.9018~−0.9750	−0.9749~0.4064	0.4064~1.2953	1.2953~2.1863	2.1863~7.1055	
Age	52.1(7.90) ^d^	52.2(8.10)	52.5(8.10)	53.1(8.40)	54.4(8.60)	<0.0001
BMI (Kg/m^2^) ^e^	24.1(2.79)	24.0(2.90)	24.0(2.76)	23.8(2.80)	23.8(2.90)	<0.0001
Energy (Kcal)	2266(578)	1903(425)	1671(372)	1496(388)	1408(364)	<0.0001
**Gender**						
Male (*n* = 55,070)	10,668(32.8) ^f^	11,060(34.0)	11,136(34.2)	10,894(33.4)	11,312(34.7)	<0.0001
Female (*n* = 107, 703)	21,857(67.2)	21,486(66.0)	21,412(65.8)	21,695(66.6)	21,253(65.3)	
**Education level**						
Elementary school	3816(11.9)	4263(13.3)	5092(15.8)	6226(19.3)	8456(26.4)	<0.0001
Middle~high school	19,029(59.3)	18,852(58.8)	18,645(57.9)	18,502(57.6)	17,802(55.5)	
College~	9246(28.8)	8982(27.9)	8415(26.2)	7411(23.1)	5832(18.1)	
**Alcohol drinking**						
Never	16,062(49.6)	15,956(49.2)	16,269(50.1)	16,755(51.6)	17,201(52.9)	<0.0001
Past	1262(3.9)	1139(3.5)	1096(3.4)	1247(3.8)	1272(3.9)	
Current	15,064(46.5)	15,330(47.3)	15,100(46.5)	14,494(44.6)	14,001(43.2)	
**Physical activity ^g^**						
Irregular	12,761(39.3)	14,274(43.9)	15,182(46.7)	16,333(50.3)	18,641(57.4)	<0.0001
Regular	19,685(60.7)	18,189(56.1)	17,301(53.3)	16,156(49.7)	13,834(42.6)	
**Income (million(s) won)**						
Less than 1	,114(7.9)	2291(8.4)	2771(10.1)	3551(12.9)	4861(17.4)	
1~2	4,806(18.1)	5082(18.5)	5409(19.7)	5926(21.6)	6760(24.2)	<0.0001
2~3	6144(23.1)	6555(23.9)	6306(23.1)	6129(22.3)	5937(21.3)	
More than 3	13,587(50.9)	13,479(49.2)	12,934(47.1)	11,865(43.2)	10,339(37.1)	
**Marital status**						
Married	29,135(90.2)	29,079(89.9)	28,805(88.9)	28,178(86.9)	27,409(84.7)	<0.0001
Unmarried/divorced	3182(9.8)	3265(10.1)	3583(11.1)	4237(13.1)	4960(15.3)	
**Smoking**						
Never	24,084(74.4)	23,772(73.3)	23,820(73.4)	24,010(73.9)	23,056(71.0)	<0.0001
Past	4427(13.7)	4782(14.8)	4783(14.8)	4570(14.0)	4591(14.1)	
Current	3857(11.9)	3864(11.9)	3842(11.8)	3905(12.1)	4821(14.9)	
**Menopause status**						
Post-menopause	11,393(56.4)	11,488(56.4)	11,795(57.9)	12,576(60.9)	13,597(65.8)	<0.0001
Pre-/peri-menopause	8801(43.6)	8870(43.6)	8570(42.1)	8082(39.1)	7073(34.2)	
**Family history of CVD**						
Negative	26,321(81.1)	26,240(80.8)	26,226(80.8)	26,430(81.4)	26,724(82.3)	<0.0001
Positive	6136(18.9)	6219(19.2)	6239(19.2)	6056(18.6)	5747(17.7)	

^a^ Dietary inflammatory index (DII^®^) score presented by the quintile at baseline, which divided the DII scores into five levels, Q1 represents the anti-inflammatory index of DII, whereas Q5 represents the highest pro-inflammatory index score of food parameters. ^b^ Jonckheere–Terpstra and Mantel–Haenszel Chi-square test was used to calculate *p* values for the trend of the continuous and categorical variables, respectively. ^c^ Range is calculated by dividing the DII score into quintiles with the same number of sample size (Q1~Q5) in the control group, subsequently confirmed the boundary value (minimum or maximum value) of the divided quintiles score and set the DII value of the CVD cases. ^d^ The data for continuous variables were presented as means (standard deviation). ^e^ BMI (Kg/m^2^) body mass index according to Asia-Pacific guidelines. ^f^ The data for categorical variables were presented as *n* (%) among all the participants. ^g^ Regularity of physical activity was measured according to whether or not the participants engaged regularly in any sports to the point of sweating.

**Table 2 nutrients-12-00588-t002:** Cox proportional hazard ratios (HRs) (95% Confidence Intervals (CIs)) for overall CVD risk by quintiles of the DII score for all participants in the KOGES_HEXA cohort, 2004~2013.

	Quintiles of DII Score ^a^					*p* ^b^
	Q1	Q2	Q3	Q4	Q5	
**All subjects**						
Person-years	253,146	243,506	241,516	238,933	228,429	
Crude HR (95% CI) ^c^	ref	1.15(0.95–1.40)	1.15(0.95–1.40)	1.46(1.21–1.76)	1.63(1.35–1.98)	0.001
Multivariate HR (95% CI) ^d^	ref	1.13(0.92–1.39)	1.08(0.86–1.35)	1.37(1.10–1.71)	1.32(1.05–1.67)	0.003
**Men**						
Person-years	81,104	81,614	82,242	79,945	80,529	
Crude HR (95% CI)	ref	1.12(0.86–1.47)	1.04(0.79–1.36)	1.28(0.98–1.67)	1.46(1.12–1.91)	0.002
Multivariate HR (95% CI)	ref	1.18(0.89–1.57)	1.10(0.81–1.49)	1.38(1.02–1.89)	1.43(1.04–1.96)	0.007
**Women**						
Person-years	172,042	161,892	159,274	158,988	147,900	
Crude HR (95% CI)	ref	1.13(0.85–1.50)	1.20(0.91–1.58)	1.56(1.20–2.03)	1.63(1.23–2.15)	<0.0001
Multivariate HR (95% CI)	ref	1.06(0.79–1.43)	1.05(0.77–1.44)	1.33(0.97–1.82)	1.19(0.85–1.67)	0.18

^a^ Dietary inflammatory index (DII^®^) score presented by the quintile at the baseline, which divided the DII scores into five levels, Q1 represents the anti-inflammatory index, whereas Q5 represents the highest pro-inflammatory index score of food parameters. ^b^
*p* value was determined using the continuous DII score. ^c^ Data are presented as hazard ratios (HRs) with correspondent 95% confidence intervals (CI). ^d^ Multivariate-adjusted for gender (for all subjects), age, smoke, alcohol consumption, physical activity, BMI, and energy intake. *p* for heterogeneity between men and women using a likelihood test = 0.64.

**Table 3 nutrients-12-00588-t003:** Cox proportional hazard ratios (HRs) (95% Confidence Intervals (CIs) for myocardial infarction (MI) and stroke risk by quintiles of DII score for all participants in the KOGES_HEXA cohort, 2004~2013.

		Quintiles of DII Score ^a^					*p* ^b^
		Q1	Q2	Q3	Q4	Q5	
**MI**							
All subjects	Crude HR (95% CI) ^c^	Ref.	1.17(0.94–1.47)	1.21(0.97–1.50)	1.37(1.10–1.70)	1.58(1.27–1.98)	<0.0001
	Multivariate HR (95% CI) ^d^	Ref.	1.14(0.90–1.43)	1.09(0.85–1.40)	1.23(0.95–1.59)	1.23(0.94–1.60)	0.05
Men	Crude HR (95% CI)	Ref.	1.06(0.78–1.44)	0.99(0.73–1.35)	1.08(0.79–1.46)	1.37(1.01–1.84)	0.04
	Multivariate HR (95% CI)	Ref.	1.07(0.78–1.48)	1.01(0.72–1.42)	1.10(0.77–1.56)	1.27(0.89–1.82)	0.13
Women	Crude HR (95% CI)	Ref.	1.26(0.91–1.74)	1.38(1.00–1.90)	1.64(1.20–2.24)	1.65(1.18–2.29)	0.0002
	Multivariate HR (95% CI)	Ref.	1.20(0.86–1.69)	1.19(0.83–1.70)	1.38(0.95–1.99)	1.18(0.79–1.75)	0.25
**Stroke**							
All subjects	Crude HR (95% CI)	Ref.	1.01(0.68–1.51)	1.01(0.68–1.50)	1.74(1.21––2.49)	1.80(1.24–2.61)	<0.0001
	Multivariate HR (95% CI)	Ref.	1.02(0.67–1.55)	1.04(0.67–1.61)	1.77(1.16–2.71)	1.55(0.99–2.44)	0.008
Men	Crude HR (95% CI)	Ref.	1.29(0.72–2.31)	1.22(0.68–2.18)	2.15(1.26–3.65)	1.90(1.09–3.32)	0.0017
	Multivariate HR (95% CI)	Ref.	1.51(0.82–2.77)	1.46(0.77–2.77)	2.66(1.44–4.94)	2.06(1.07–3.98)	0.003
Women	Crude HR (95% CI)	Ref.	0.77(0.44–1.35)	0.78(0.44–1.37)	1.34(0.81–2.20)	1.57(0.94–2.61)	0.06
	Multivariate HR (95% CI)	Ref.	0.68(0.37–1.24)	0.74(0.40–1.37)	1.15(0.63–2.09)	1.17(0.62–2.20)	0.59

^a^ Dietary inflammatory index (DII^®^) score presented by the quintile at the baseline, which divided the DII scores into five levels, Q1 represents the anti-inflammatory index, whereas Q5 represents the highest pro-inflammatory index score of food parameters. ^b^
*p* value was determined using the continuous DII score. ^c^ Data are presented as hazard ratios (HRs) with corresponding 95% confidence intervals (CI). ^d^ Multivariate-adjusted for gender (for all subjects), age, smoke, alcohol consumption, physical activity, BMI, and energy intake.

**Table 4 nutrients-12-00588-t004:** Cox proportional HRs (95% Confidence Intervals (CIs)) for CVD risk by quintiles of DII score after stratification by physical activity, smoking, and BMI.

Strata		Quintiles of Dietary Inflammatory Index (DII) ^a^					*p* Value ^c^
		Q1	Q2 ^b^	Q3	Q4	Q5	
**Physical activity ^d^**							
All subjects	Irregular	Ref.	1.32(0.94–1.85)	1.25(0.88–1.77)	1.60(1.12–2.26)	1.42(0.99–2.04)	0.03
	Regular	Ref.	1.03(0.79–1.33)	0.98(0.74–1.30)	1.23(0.92–1.64)	1.28(0.94–1.74)	0.05
Men	Irregular	Ref.	1.65(0.98–2.78)	1.37(0.79–2.36)	1.69(0.97–2.92)	1.80(1.03–3.12)	0.03
	Regular	Ref.	1.04(0.74–1.47)	1.03(0.71–1.49)	1.33(0.91–1.94)	1.30(0.86–1.94)	0.09
Women	Irregular	Ref.	1.11(0.71–1.72)	1.19(0.75–1.87)	1.56(0.99–2.46)	1.17(0.72–1.92)	0.37
	Regular	Ref.	1.01(0.67–1.50)	0.90(0.59–1.39)	1.08(0.69–1.70)	1.24(0.78–1.98)	0.36
**BMI (Kg/m^2^) ^e^**							
All subjects	Normal (≤22.9)	Ref.	1.18 (0.79–1.75)	1.08(0.71–1.65)	1.27(0.82–1.97)	1.29(0.82–2.03)	0.31
	Overweight (≤24.9)	Ref.	1.09(0.74–1.59)	1.32(0.89–1.95)	1.64(1.10–2.45)	1.35(0.88–2.08)	0.02
	Obese (≥25)	Ref.	1.14(0.84–1.55)	0.94(0.67–1.31)	1.26(0.90–1.76)	1.34(0.95–1.90)	0.07
Men	Normal (≤22.9)	Ref.	1.85(1.02–3.33)	1.05(0.53–2.07)	1.72(0.89–3.31)	1.33(0.66–2.68)	0.25
	Overweight (≤24.9)	Ref.	0.95(0.55–1.62)	1.32(0.78–2.25)	1.10(0.62–1.95)	1.15(0.63–2.06)	0.36
	Obese (≥25)	Ref.	1.05(0.70–1.60)	0.97(0.62–1.51)	1.45(0.93–2.26)	1.77(1.13–2.76)	0.01
Women	Normal (≤22.9)	Ref.	0.75(0.43–1.31)	1.09(0.64–1.88)	0.98(0.55–1.77)	1.26(0.69–2.30)	0.87
	Overweight (≤24.9)	Ref.	1.19(0.68–2.10)	1.20(0.66–2.18)	2.38(1.34–4.21)	1.65(0.87–3.12)	0.01
	Obese (≥25)	Ref.	1.23(0.78–1.93)	0.91(0.55–1.50)	1.01(0.60–1.69)	0.88(0.51–1.52)	0.76
**Smoking**							
All subjects	No	Ref.	1.08(0.83–1.40)	0.97(0.74–1.29)	1.33(1.0–1.76)	1.21(0.89–1.63)	0.16
	Yes	Ref.	1.21(0.87–1.69)	1.24(0.88–1.76)	1.41(0.99–2.00)	1.52(1.05–2.18)	0.004
Men	No	Ref.	1.02(0.61–1.71)	0.70(0.39–1.25)	1.32(0.75–2.32)	1.07(0.57–1.98)	0.95
	Yes	Ref.	1.27(0.90–1.78)	1.20(0.90–1.84)	1.41(0.97–2.03)	1.60(1.10–2.33)	0.002
Women	No	Ref.	1.10(0.81–1.48)	1.07(0.78–1.47)	1.32(0.95–1.83)	1.23(0.87–1.74)	0.12
	Yes	Ref.	0.58(0.12–2.71)	0.78(0.17–3.53)	1.63(0.38–6.97)	0.66(0.12–3.44)	0.54

^a^ Dietary inflammatory index (DII^®^) score presented by the quintile at the baseline, which divided the DII scores into five levels, Q1 represents the anti-inflammatory index, whereas Q5 represents the highest pro-inflammatory index score of food parameters. ^b^ Data are presented as hazard ratios (HRs) with corresponding 95% confidence intervals (CI). ^c^
*p* value was determined using the continuous DII score. ^d^ Physical activity-regular was considered if the subjects performed any activity to the point of sweating. ^e^ BMI: body mass index was measured according to Asia-Pacific guidelines. Multivariate-adjusted for gender (for all participants), age, smoke, alcohol consumption, physical activity, BMI, and energy intake.

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
