# Peer review of "Positive Association of Dietary Inflammatory Index with Incidence of Cardiovascular Disease: Findings from a Korean Population-Based Prospective Study"

_nutrients, 2020, doi:10.3390/nu12020588_

Round 1

Reviewer 1 Report

In this prospective cohort study, the authors found the association between dietary inflammatory index (DII) and the incidence of cardiovascular diseases, in particular stroke. This association was greater in man, smokers, obese and physically inactive individuals.

I found this work interesting and clear. Number of individuals is large and the follow-up period appropriate. Statistical analyses were correctly performed. Limitations of the study were also aknowledged.

I have only minor comments:

-Stroke and myocardial infarction present a multifactorial pathogenesis, which includes also genetics besides lifestyle and diet. Episodes of stroke and myocardial infarction are recurrent across generations. Individuals with lower DII but with a family history of CVDs are more predisposed to develop stroke when compared to individuals without a family history of CVDs with a higher DII? Please discuss this aspect, even with data collected.

-Are the blood inflammatory parameters (baseline/follow-up) of these subjects unavailable?

-Which sub-type of stroke was included in the analysis? Ischemic stroke, haemorrhagic, transient-ischemic attack. All three? Please clarify.

Author Response

Response

We would like to extend our gratitude towards the Editors and Reviewers for their time and valuable comments on our manuscript.

Reviewer 1.

  1. Stroke and myocardial infarction present a multifactorial pathogenesis, which includes also genetics besides lifestyle and diet. Episodes of stroke and myocardial infarction are recurrent across generations. Individuals with lower DII but with a family history of CVDs are more predisposed to develop stroke when compared to individuals without a family history of CVDs with a higher DII? Please discuss this aspect, even with data collected.

Response: Stroke and myocardial infarction present a multifactorial pathogenesis, which also includes genetics in addition to lifestyle and diet. Episodes of stroke and myocardial infarction may occur across generations. However, in the present study, compared to individuals without a family history of CVDs with a higher DII, individuals with lower DII but with a family history of CVDs were not more likely to develop stroke. We added it to the text. (Line 262~266)

  1. Are the blood inflammatory parameters (baseline/follow-up) of these subjects unavailable?

Response: We have blood CRP data in this cohort, analyses of CRP in relation to CVD, and other selected diseases are now on-going.

  1. Which sub-type of stroke was included in the analysis? Ischemic stroke, haemorrhagic, transient-ischemic attack. All three? Please clarify.

Response: Stroke, myocardial infarction, angina, and transient-ischemic attack are defined as CVD, and all types of CVD were included in the present study. (Line 104-105, Figure 1)

Additionally, we corrected some simple typos.

Reviewer 2 Report

The authors present an interesting study whose aim was to prospectively investigate the association between the inflammatory potential of the diet, as measured by the DII, and the incidence of overall CVD and its subtypes MI and strokes in a large population-based Korean Genome and Epidemiology Study Health Examination (KoGES_HEXA) cohort.

The manuscript is properly structured and drafted.

The Introduction section adequately contextualizes the problema. The authors describe, according to scientific evidence, the utility of dietary inflammatory index (DII®) as a tool to assess the inflammatory potential of an individual's diet.

The Research design and methods section adequately describes the inclusion criteria, and the final sample size. However, the authors should improve the following aspects:

- The authors should better describe the conditions for carrying out biochemical determinations, collection of nutritional variables during the follow-up medical exams (physical place, specific protocols, trained personnel, ...).

- What was the tool used to assess physical activity? The authors indicate: "Regularity of physical activity was measured according to whether or not participants contributed regularly in any sports to the point of sweating". According to this statement, was the "Physical activity" variable self-reported by the participants?

The Results section clearly describes the main findings of the study. The Tables are pertinent and improve the compression of the contents.

The Discussion section is consistent with the results. The authors discuss adequately the results obtained, comparing with studies of similar characteristics.

In Conclusion section, authors should better indicate the utility of these results in the clinical setting with adults population.

The bibliography consulted is pertinent.

Author Response

Response

We would like to extend our gratitude towards the Editors and Reviewers for their time and valuable comments on our manuscript.

Reviewer 2.

  1. The authors should better describe the conditions for carrying out biochemical determinations, collection of nutritional variables during the follow-up medical exams (physical place, specific protocols, trained personnel, ...).

Response: The details of the data collection and processing for the baseline recruitment and follow-up examinations was explained in the Material and Methods section. (Line 116~127)

  1. What was the tool used to assess physical activity? The authors indicate: "Regularity of physical activity was measured according to whether or not participants contributed regularly in any sports to the point of sweating". According to this statement, was the "Physical activity" variable self-reported by the participants?

Response: Yes, we used a structured questionnaire, and added its description in the revised text. (Line 116~127)

  1. In Conclusion section, authors should better indicate the utility of these results in the clinical setting with adults population.

Response: We have added text to the Conclusion. (Line 340~343)

Additionally, we corrected some simple typos.